**Subject Area:**
immunology/microbiology

IL-17, viral infections, antiviral immunity, inflammation, immunopathogenesis

**Authors for correspondence:**
Wen-Tao Ma
e-mail: mawentao@nwafu.edu.cn
De-Kun Chen
e-mail: cdk@nwafu.edu.cn

# The protective and pathogenic roles of IL-17 in viral infections: friend or foe?

Wen-Tao Ma, Xiao-Ting Yao, Qun Peng and De-Kun Chen

College of Veterinary Medicine, Northwest A&F University, Yangling 712100, Shaanxi Province, People's Republic of China

(ID) W-TM, 0000-0002-4747-1489

Viral infections cause substantial human morbidity and mortality, and are a significant health burden worldwide. Following a viral infection, the host may initiate complex antiviral immune responses to antagonize viral invasion and replication. However, proinflammatory antiviral immune responses pose a great threat to the host if not properly held in check. Interleukin (IL)-17 is a pleiotropic cytokine participating in a variety of physiological and pathophysiological conditions, including tissue integrity maintenance, cancer progression, autoimmune disease development and, more intriguingly, infectious diseases. Abundant evidence suggests that while IL-17 plays a crucial role in enhancing effective antiviral immune responses, it may also promote and exacerbate virus-induced illnesses. Accumulated experimental and clinical evidence has broadened our understanding of the seemingly paradoxical role of IL-17 in viral infections and suggests that IL-17-targeted immunotherapy may be a promising therapeutic option. Herein, we summarize current knowledge regarding the protective and pathogenic roles of IL-17 in viral infections, with emphasis on underlying mechanisms. The various and critical roles of IL-17 in viral infections necessitate the development of therapeutic strategies that are uniquely tailored to both the infectious agent and the infection environment.

## 1. Introduction

Viral infections are common causes of both chronic and acute tissue pathology that create a significant health burden worldwide. For example, human papillomaviruses (HPV) are the causative agents of epithelial hyperplasia of the skin and genital tract [1]. Persistent infection of HPV can result in malignant lesions, the most common being cervical cancer, which had caused an estimated 0.6 million new cases in 2018 and contributed to more than 0.3 billion deaths [2]. Hepatitis B virus (HBV) and hepatitis C virus (HCV) are the major causes of hepatitis and can progress to liver fibrosis, cirrhosis and eventually liver cancer (the seventh most common human cancer worldwide) [2]. Human immunodeficiency virus (HIV) infection can cause acquired immune deficiency syndrome, which accounts for approximately 1.5 million deaths per year worldwide [3]. Unfortunately, for many viral illnesses, there are no effective and specific treatments. Therefore, a better understanding of how viral infections are controlled or promoted by the host may shed light on better clinical care approaches and the development of novel therapeutic strategies.

An expansive history of evidence has revealed the essential role of the host immune system in preventing viral infections [4–6]. The initial sensing of an invading virus by pattern recognition receptors of the host innate immune system induces the production of interferons (IFN) and other proinflammatory cytokines as a part of the early host antiviral response phase. Afterwards, both the activation of cytotoxic T lymphocytes (CTLs) and B-cell production of neutralizing antibodies ultimately mounts an effective and specific antiviral

royalsocietypublishing.org/journal/rsob   Open Biol. 9: 190109

response for optimal viral clearance. However, despite the essential need for viral control and clearance, the intensity of the antiviral immune response must be delicately regulated to avoid excessive inflammation and associated tissue damage during both acute and chronic viral infections [6,7].

Interleukin (IL)-17 was discovered in the 1990s and has since emerged as a remarkably pleiotropic cytokine that contributes in unique ways to the host immune response. IL-17 can be produced by a wide range of immune cell populations, such as Th17 cells [8], CD8$^+$ T cells [9], $\gamma\delta$T cells [10], natural killer (NK) cells [11], natural killer T (NKT) cells [12], mast cells [13], neutrophils [14] and group 3 innate lymphoid cells (ILC3) [15]. IL-17 plays a key role in the maintenance of tissue integrity and the generation of protective immune responses to infectious microorganisms, especially at epithelial barrier sites [16]. The proinflammatory properties of IL-17 also make it a crucial mediator of inflammation and immunopathology [16]. The surprisingly diverse functions of IL-17 have made it among the more favourable immunotherapeutic target candidates for the treatment of a wide range of diseases, including cancers [17], autoimmune diseases [18] and infectious diseases [19]. Research is also being done on IL-17 in the regulation of viral infections, where it plays varied and crucial roles. Intriguingly, a body of literature indicates that while IL-17 is a critical player in host defence by substantially suppressing viral infections, it has also been strongly implicated in the promotion of viral infection and related illness. Herein, we focus on the beneficial and detrimental effects of IL-17 during viral infections, as well as infection-induced tissue pathology. We also focus on elucidating relevant mechanisms underlying these observations. In addition, we further highlight the translational significance of IL-17-targeted immunotherapies as promising therapeutic options for the treatment of viral infections and resulting tissue pathology.

## 2. IL-17 hinders viral infections and limits viral infection-related illness2

### 2.1. Mechanisms by which IL-17 hinders viral infections

#### 2.1.1. Enhancing Th1 immune responses

It is well known that Th1 cells are crucial for the development of antiviral immunity against genital infection by herpes simplex virus (HSV). In brief, either CD4$^+$ T-cell deletion or IFN-$\gamma$ blockade results in significantly compromised protection against lethal HSV challenge [20], while enhanced Th1 immune responses or Th1-cytokine administration reduces both the morbidity and mortality of mice following HSV challenge [21,22]. The seminal work of Kaushic and colleagues demonstrated that IL-17 is crucial for the generation of efficient Th1 immune responses in mice following intravaginal HSV-2 infection. In their observations, IL-17 is primarily secreted by vaginal Th17 cells, which are induced through IL-1$\beta$ produced by CD11c$^+$ dendritic cells (DCs) [23]. IL-17-deficient mice failed to mount a protective Th1 immune response and were more prone to genital HSV-2 challenge [23]. Interestingly, it appears that the promotion of antiviral Th1 cell immunity by IL-17 is more pronounced following HSV-2 reexposure or rechallenge in mice [24]. In IL-17A-knockout mice, the lack of IL-17A production resulted

in higher viral shedding, more severe morbidity and mortality, and significantly compromised Th1 immune responses, with subsequently fewer IFN-$\gamma$-expressing CD4$^+$ tissue-resident memory T cells observed in the female genital tract [24]. In comparison, IL-17 did not play a significant role during primary HSV-2 infection [24].

Collected works of data suggest IL-17 is crucial for enhancing Th1 immune responses against HSV-2 infection in the vaginal tract, with an effect that is more evident following virus rechallenge. It is of particular interest to consider inducing IL-17 production during HSV-2 vaccine administration. However, further work is still needed to determine whether IL-17 is directly responsible for facilitating Th1 immune responses following HSV-2 infection and to reveal how IL-17 regulates the generation of tissue-memory CD4$^+$ T cells in the vaginal tract.

#### 2.1.2. Promoting cytotoxic T-cell activity

CD8$^+$ cytotoxic T cells are pivotal components of antiviral immunity and play a crucial role in mediating virus clearance [4]. CD8$^+$ T cells eliminate invading viruses via two primary mechanisms: direct cytotoxicity of infected cells and via production of multiple cytokines to induce local or systemic antiviral responses [25,26]. In scenarios such as tumour development [27], cigarette smoke-induced emphysema [28] and parasite infection [29], IL-17 has been shown to play a critical role in the activation and survival of cytotoxic CD8$^+$ T cells. IL-17 also exhibits similar CD8$^+$ T-cell-modulating functions during various viral infections, as discussed below.

Using a murine model of West Nile virus infection, Acharya et al. [30] found that IL-17-deficient mice exhibited a higher viral burden, as well as a lower survival rate. Both of these responses were paired with reduced CD8$^+$ T-cell cytotoxicity characterized by a lower expression of cytotoxic-mediator genes (i.e. Perforin-1, Granzymes and FasL) [30]. Of note, treatment with recombinant IL-17 significantly recovered the cytotoxicity of CD8$^+$ T cells from IL-17-deficient mice [30]. It appears that the effect of IL-17 on CD8$^+$ T cells is direct, as the expression of cytotoxic-mediator genes of sorted splenic CD8$^+$ T cells could be directly induced by addition of recombinant IL-17 in vitro [30]. Consistent with this finding, Hou et al. [31,32] found that IL-17 was crucial for priming hepatic CD8$^+$ T cells since IL-17 neutralization or IL-17RA-knockout resulted in significantly decreased CTL counts and compromised CTL effector functions in adenovirus-infected mice. In their subsequent work, they also showed that IL-17 production was IL-7 dependent, as IL-7R blockade resulted in a markedly decreased number of IL-17-producing cells following viral infection [31]. In addition, potent type I IFN signalling induced by adenovirus infection was primarily responsible for hepatic IL-7 induction [31]. IL-17 was also shown to be partly responsible for the clustering and activation of CD8$^+$ T cells in the spleen following vaccinia virus infection, thereby contributing to host defence against viral infection [33].

In certain viral infections, IL-17 can also promote CD8$^+$ T-cell cytotoxicity to affect viral clearance. In some cases, IL-17-induced activation of CD8$^+$ T cells has been shown to coincide with increased CD8$^+$ T-cell number [30,31]. Thus, it is of particular interest to evaluate whether IL-17 can facilitate CD8$^+$ T-cell recruitment or promote their survival in this scenario. Although IL-17 can directly activate CD8$^+$ T cells, it

royalsocietypublishing.org/journal/rsob    Open Biol. 9: 190109

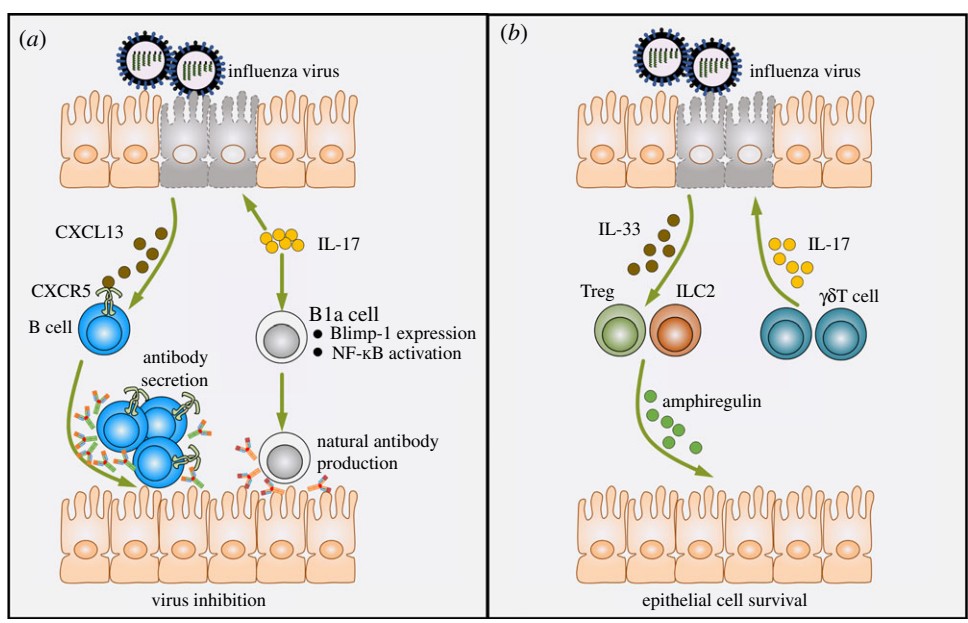

**Figure 1.** IL-17 can enhance antiviral B-cell activities and promote the survival of virus-infected cells during influenza virus infection. Upon influenza virus infection, (*a*) IL-17 induces the secretion of CXCL13 from necrotic lung epithelial cells. CXCL13 is a potent chemoattractant for CXCR5$^+$ B cells, which migrate into the lung and produce abundant neutralizing antibodies to suppress influenza virus infection. In addition, IL-17 can induce Blimp-1 expression and NF-κB activation in B1a cells, resulting in natural IgM antibody production and the suppression of viral infection. (*b*) γδT-cell-derived IL-17 stimulates IL-33 secretion by necrotic lung epithelial cells. IL-33 facilitates the induction and infiltration of ILC2s and Tregs, which secrete amphiregulin to promote the growth of lung epithelial cells. CXCL13, CXC-chemokine ligand 13; CXCR5, CXC-chemokine receptor 5; Treg, regulatory T cell; ILC2, type 2 innate lymphoid cell.

has yet to be demonstrated whether IL-17 can indirectly induce CD8$^+$ T-cell activation via effects on the antigen-presenting cells, which essentially participate in T-cell activation and can be stimulated by IL-17, as has been indicated in many studies [34–37].

Learning to beneficially control the regulation of viral-specific CD8$^+$ T-cell activation to promote viral clearance is vital information. It has been observed that a higher Th17 percentage is associated with increased T-cell proliferation and more potent simian immunodeficiency virus (SIV)-specific CTL responses in SIV-infected rhesus macaques [38,39]. In this model, prior transfer of Th17 cells was shown to enhance the induction of antigen-specific CTLs following vaccination with an adenovirus vector [40]. Thus, the development of IL-17-based therapeutic strategies and vaccines will be of particular future importance.

### 2.1.3. Modulating antiviral B-cell activities

Accumulating evidence reveals that B cells are critically involved in the host's control of multiple viruses [41]. The protective role of B cells during viral infection is largely mediated through an effective humoral immune response that includes both the production of natural antibodies at infection onset and isotype-switched virus-specific antibodies at a later phase of infection [42–45]. Although several studies have investigated the role of IL-17 in the induction and maintenance of effective humoral immunity in various settings [46–49], studies regarding the effect of IL-17 in modulating antiviral B-cell activities have been very limited. In 2011, the pioneering work of Wang *et al.* [50] showed that following H5N1 influenza virus infection, IL-17 is critical for recruitment of B cells to the lung. This phenomenon is partly mediated through the induction of CXC-chemokine ligand (CXCL)13, a potent chemoattractant for CXC-chemokine receptor (CXCR)5-expressing B cells [50] (figure 1*a*). Further

work by the same group revealed an intriguing function of IL-17 to promote the migration and differentiation of B1a cells, a subset of B cells that are the major source of natural immunoglobulin (Ig)M [42,51]. During H1N1 influenza infection, the deficiency of IL-17 was shown to result in significantly compromised B1a cell-derived antibody production in the lung that coincided with defective viral clearance. Mechanistically, IL-17 was shown to be essential for the induction of B lymphocyte-induced maturation protein (Blimp)-1 expression and nuclear factor kappa-light-chain-enhancer of activated B cells (NF-κB) activation of pulmonary B1a cells. This response leads to the plasmacytic differentiation of these cells and eventually the production of natural IgM [42] (figure 1*a*).

The collective published evidence supports a beneficial role of IL-17 in modulating activities of antiviral B cells. However, future studies are urgently needed to verify whether these conclusions also apply to human cases. In addition to inducing the chemotaxis of conventional B cells, confirming the effects of IL-17 on B-cell maturation and antibody secretion profiles, which have been shown in studies using several other models, also warrant further evaluation.

### 2.1.4. Inducing protective inflammatory responses

Effector CD8$^+$ T cells can be classified into at least two functional subsets: IFN-γ-producing type 1 CD8$^+$ T (Tc1) cells that display potent cytotoxic capacities, and IL-17-producing type 17 CD8$^+$ T (Tc17) cells with low killing activity [52]. While the mechanisms for overcoming multiple viral infections by Tc1 cells have been clearly demonstrated, mechanisms for the suppression of viral infections by Tc17 cells have not been fully elucidated. In response to being challenged with lethal doses of influenza infection, the number of Tc17 cells in the lung increased sharply and was shown to play a host-beneficial role, since the neutralization of IL-17 resulted in higher morbidity and mortality of the infected mice [53].

However, the majority of Tc17 cells displayed no cytolytic activity, while the protection conferred by these cells coincided with an influx of neutrophils into the lung [53]. Consistent with this observation, following the infection of HSV in mice, IL-17 appears to be responsible for reduced viral load partly by strengthening inflammatory responses that promote neutrophil migration into the lung [54]. Following vaccinia virus infection, adoptively transferred Tc17 cells can acquire strong cytotoxic potential *in vivo*, as shown by their increased FasL expression levels that correlate with effective viral clearance [55].

Together, the data suggest that IL-17 can mediate antiviral inflammatory responses through the induction of neutrophil migration, although very few studies have been reported with regard to this phenomenon. Therefore, additional work is needed to clarify how IL-17 orchestrates host protective inflammatory responses during viral infections.

## 2.2. Mechanisms by which IL-17 limits viral infection-induced organ pathology

### 2.2.1. Contributing to maintenance of tissue integrity

It has long been recognized that IL-17 plays a central role in the maintenance of tissue integrity, repair and regeneration. For example, in colonic tissue, $\gamma\delta$T-cell-derived IL-17A contributes to the regulation of tight junction protein occludin in response to epithelial injury. This in turn limits abnormal tissue permeability and maintains epithelial integrity [56]. In addition, the role of IL-17 seems particularly relevant to liver and bone regeneration, since IL-17 both functions directly as a mitogen and promotes the generation of pro-regenerative leucocytes upon tissue injury [57,58]. Consistent with these findings, several studies suggest the potential involvement of IL-17 in the maintenance of tissue integrity after viral infections.

During chronic SIV or HIV infection, there is a substantial insult to the gut mucosa, characterized by a progressive depletion of $CD4^+$ T cells and a concomitant breakdown of the epithelial barrier of the gastrointestinal (GI) tract [59,60]. These changes result in increased microbial translocation from the GI tract, further contributing to the systemic dissemination of microbial constituents, pathological immune activation and progressive disease development [61]. In an effort to characterize immunological alterations in the GI tract during SIV infection, Klatt *et al.* [62] found that damaged epithelial integrity was associated with markedly reduced IL-17 production. Further investigation revealed a loss of IL-17-producing cells due to the depletion of $CD103^+$ DCs, which have been shown to both express genes favourable to IL-17 production and induce the differentiation of *IL-17A*- and *RORc*-expressing cells upon coculture with naive T cells *in vitro* [62]. Consistent with this finding, Reeves *et al.* [63] demonstrated that a subset of mucosal $NKp44^+$ NK cells (which secrete IL-17A and IL-22 and are involved in the maintenance of gut integrity) were depleted and exhibited an aberrant functional profile of elevated IFN-$\gamma$ production and decreased IL-17A production. This phenomenon is believed to be caused by increased indoleamine 2,3-dioxygenase 1 production after SIV infection [63].

Taken together, IL-17 appears to protect tissue integrity during viral infections. Thus, it would be of interest to determine whether the addition of recombinant IL-17 would be effective in treating tissue damage during the infection of certain viruses.

### 2.2.2. Inhibiting detrimental inflammations

Although IL-17 is well recognized as an active inducer of inflammation, recent studies also indicate that it is also potentially involved in the suppression of detrimental tissue inflammations during viral infections. Huang *et al.* [64] found that neonates failed to develop an IL-17 response during respiratory syncytial virus (RSV) infection and exhibited more severe lung inflammation and pathology than adult mice. To examine the function of IL-17 during RSV infection, they found that exogenous IL-17 administration to RSV-challenged neonatal mice led to a decreased lung inflammation profile, while IL-17 neutralization in adults caused increased inflammation and airway mucus [64]. Consistent with this finding, in SIV-infected rhesus macaques, the percentage of mucosal Th17 cells is negatively correlated with plasma SIV levels, with Th1 cell number predominating over that of Th17 cells in highly viraemic animals [40]. Moreover, during and after antiretroviral therapy, Th17 cell function and frequency were negatively associated with both systemic inflammation and virus persistence [65]. Similar findings were obtained for Tc17 cells in HIV patients receiving antiretroviral therapies [66]. In these patients, persistent immune activation was still evident, despite effective viral suppression, and was associated with a low frequency of $CD161^+$ Tc17 cells exhibiting severely compromised IL-17 secretory ability. Importantly, this Tc17 cell dysfunction could be partially recovered after anti-inflammatory agent treatment [66].

The discussed body of evidence implies that IL-17 may participate in suppressing local or systemic inflammations during viral infections. This issue has not been explored extensively and very little is known about how IL-17 negatively modulates inflammatory immune responses. This is an important topic to explore, especially considering the inflammatory responses during certain types of viral infections are usually fatal. Therefore, further investigations to elucidate the role and mechanism(s) of IL-17 action in suppressing inflammatory immune responses during viral infections are clearly needed.

### 2.2.3. Mediating protective immune responses

Although rapid type 1 immune responses contribute to efficient influenza virus clearance, ILC2 activation and the correspondingly induced type 2 immune responses are indispensable for lung repair and homeostatic restoration [67,68]. Similarly, regulatory T (Treg) cells have been suggested to play a protective role in promoting lung tissue integrity and repair through IL-33-mediated production of the growth factor amphiregulin [69–72]. Guo *et al.* [73] reported that IL-17-producing $\gamma\delta$T cells functioned as crucial immune regulators following the infection of neonatal mice with influenza. Although the activation of this cell subset did not affect viral clearance or IFN-$\gamma$ production, IL-17 secretion stimulated IL-33 production by lung epithelial cells. In turn, IL-33 production further facilitated lung infiltration of ILC2s and Treg cells, both of which are important sources of amphiregulin that is necessary for tissue repair (figure 1*b*). Importantly, neonates lacking IL-17-producing $\gamma\delta$T cells or

IL-33 production showed more serious morbidity and mortality. This response is of particular interest due to previously observed influenza-affected children profiles. In these children, the concentrations of IL-17A, IL-33 and amphiregulin in the nasal aspirates were shown to correlate with one another, with higher IL-17 levels correlating with better clinical prognosis [73].

Overall, an IL-17-orchestrated protective immune response is essential for lung repair following influenza infection in neonates. This evidence may suggest that the IL-17/IL-33/amphiregulin axis may serve as a potential therapeutic target that is specifically important in cases of influenza infection in neonates. However, further investigations, including additional prospective cohort studies, are needed to assess the applicability of this concept towards the development of neonatal influenza treatments.

# 3. IL-17 promotes viral infections and mediates viral infection-induced pathology

Despite its protective roles in the suppression of viral infections and infection-induced organ pathology, IL-17 has also been strongly associated with promotion of viral infections and tissue pathology. The exploration of mechanisms underlying this phenomenon is a currently active research area.

## 3.1. Mechanisms by which IL-17 promotes viral infections

### 3.1.1. Antagonizing antiviral Th1 or CTL responses

As discussed above, while antiviral Th1 and CTL responses are essential for eliminating invading viruses, a plethora of evidence shows that IL-17 can antagonize these instrumental immune responses. For example, following Theiler's murine encephalomyelitis virus infection in mice, IL-17 can exert direct inhibitory effects on cytolytic ability of virus-specific CTL responses both *in vitro* and *in vivo* [74,75]. Likewise, following RSV or coxsackievirus infection, the elevation of IL-17 is associated with more compromised Th1 or CTL responses, while the neutralization of IL-17 always results in correspondingly more vigorous antiviral immunity [76,77]. Mechanistically, the suppression of antiviral immunity by IL-17 is achieved through direct suppression of *IFN-γ*, *T-bet* and *eomesodermin* expressions in T cells [76,77], as well as interference of the interaction between CTLs and virus epitope-bearing target cells through the blocking of Fas-FasL signals [75].

Taken together, these data suggest that IL-17 participates in suppressing antiviral Th1 or CTL responses following viral infection, thereby fostering viral persistence and the concomitant pathogenesis. It is of particular interest to test whether these findings can be translated into effective clinical therapeutics in the future, considering the restored antiviral immunity and a correspondingly decreased viral load observed after IL-17 neutralization.

### 3.1.2. Enhancing the survival of virus-infected cells

Theiler's murine encephalomyelitis virus-induced chronic demyelinating disease displays symptoms similar to those of progressive multiple sclerosis in humans. In this murine model, persistent viral infection is always accompanied by vigorous production of IL-17. In their seminal work investigating the role of IL-17 using this model, Hou *et al.* [74,75] found that IL-17 plays a vital role in protecting permissive cells from virus-induced apoptosis, as well as through the desensitization of CTL killing of these cells. The underlying mechanism involves the induction of antiapoptotic proteins (i.e. Bcl-xL and Bcl-2) by IL-17, consistent with the reported growth-promoting effect of this cytokine in several other models [78,79]. Nevertheless, in this model, the development of pathogenic IL-17-producing cells is driven by the production of excessively high levels of IL-6 following viral infection. The concerted action of these two cytokines suppresses apoptosis of the permissive cells more efficiently than either cytokine alone to ultimately support viral persistence.

However, it should be noted that the promotion of virus permissive cells by IL-17 does not always promote viral infections. Peng *et al.* [80] observed that during HSV-2 infection, keratinocytes were the major source of IL-17c (another IL-17 family member mainly produced by epithelial cells) that was associated with a correspondingly increased expression of IL-17c receptor on skin nerve fibres. Mechanistically, IL-17c provided survival signals to stimulate growth and branching of peripheral neurons during HSV-2 infection, explaining the phenomenon that peripheral nerve destruction and sensory anaesthesia rarely happen during HSV infection [80].

### 3.1.3. Promoting virus replication

A large body of evidence has shown that Th17 cells are susceptible to and permissive for HIV/SIV infection. For example, in the presence of Th17-polarizing cytokines IL-1β, IL-6, IL-23 and TGF-β, HIV infection of T cells is significantly increased *in vitro* [39,55]. Meanwhile, during acute HIV/SIV infection *in vivo*, Th17 cells are rapidly and preferentially depleted from gut-associated lymphoid tissues as a typical form of HIV pathology [29,55]. The mechanisms underlying the preferential infection of Th17 cells by HIV/SIV may involve a higher expression of HIV-binding proteins CXCR4, $\alpha 4\beta 7$ integrin, C–C chemokine receptor type (CCR)5 and CD4, as well as a lower expression of HIV-inhibitory chemokine macrophage inflammatory protein (MIP)-1β by these cells [29,78]. Recently, the work of Christensen-Quick and colleagues [39] has shed more light on this phenomenon. In their observations, Th17 cells produce more viral capsid protein following HIV infection and could even support infection with replication-defective HIV vectors possessing pseudotype envelopes [39]. In addition, Th17 cells were shown to express lower levels of HIV-suppressive RNase 6, a key determinant for enhanced intracellular viral replication and production in these cells [39].

In summary, Th17 cells are susceptible to HIV entry and support intracellular viral replication and production upon successful virus internalization. However, this conclusion is primarily based on data from peripheral blood-isolated

royalsocietypublishing.org/journal/rsob    Open Biol. **9**: 190109

T cells *in vitro*. Given that HIV persistence generally occurs in gut-associated lymphoid tissues, a unique compartment harbouring the majority of the body's lymphocytes, we must point out that this environment differs greatly from peripheral blood with regard to cell percentages, phenotypes, activation states and effector functions [52–54]. Thus, it still remains unclear whether Th17 cells from gut-associated lymphoid tissues are also permissive for HIV infection and whether IL-17 plays a key role during this process.

## 3.2. Mechanisms by which IL-17 contributes to tissue pathology during viral infections

### 3.2.1. Inducing excessive neutrophil migration and activation

Neutrophils are one of the major players during inflammation and well recognized as the first cells to be recruited to an infection site [81]. Data from clinical and experimental settings have shown that neutrophils play crucial roles in eliminating invading pathogens [82–84]. However, mounting evidence also shows that if neutrophils are not controlled properly, they can mediate immunopathology in both acute and chronic diseases [85–87]. This pathological function appears to be exerted via several mechanisms, including the excessive production of proinflammatory cytokines, the release of proteases and oxidants and the uncontrolled formation of neutrophil extracellular traps [88].

Extensive evidence suggests a critical role of IL-17 in triggering neutrophil accumulation and/or activation that subsequently lead to tissue immunopathology during viral infection, as observed during several types of airway infection. During RSV infection, it has been observed that increased IL-17 level is usually associated with more severe tissue pathology and higher mortality after infection [89,90]. Mechanistically, IL-17 can boost the production of several chemokines (i.e. IL-8 and CXCL1) by airway epithelial cells that lead to the excessive accumulation of neutrophils, thereby generating a strong inflammatory response and thus leading to airway tissue injury [77,90,91]. Accordingly, *IL-17* gene deletion or IL-17 protein blockade can significantly decrease the extent of local inflammation and ameliorate tissue damage [77,92,93]. Consistent with these findings, a significant increase in IL-17 level has been observed during human influenza infection, as well as in relevant mouse models [94]. Intriguingly, mucosal pre-exposure to Th17-inducing adjuvants before influenza infection results in increased infiltration of neutrophils, more severe lung inflammation and increased morbidity upon subsequent infection [95]. In addition, IL-17RA, which is a common receptor of IL-17A and IL-17F, is critical for neutrophil migration and lung injury after influenza infection [96].

IL-17 also mediates tissue injury during infections by several viruses other than respiratory viruses. During murine systemic infection by HSV, Stout-Delgado *et al.* [12] found that the increased morbidity and mortality of aged mice was caused by a rapid increase in IL-17 level (primarily produced by hepatic NKT cells) during the infection. This stimulated the activation of hepatic neutrophils that subsequently contributed to hepatocyte necrosis and virus-induced death [12]. In addition, serum IL-17 levels have been observed to be significantly higher in patients infected by dengue virus. Notably, IL-17 appears to be involved in more severe cases

of infection, although the exact role of IL-17 in the pathogenesis of dengue virus infection remains unknown [97].This finding was corroborated by an independent investigation, which confirmed that the increased levels of IL-17, mostly produced by hepatic $\gamma\delta$T cells, were accompanied by neutrophil accumulation in the liver that contributed to disease progression during dengue virus infection of mice [98]. Importantly, IL-17R deletion or IL-17-neutralizing monoclonal antibody treatment resulted in substantial protection of the infected mice from virus-induced tissue pathology and mortality [98]. Consistent with these findings, increased IL-17-associated tissue damage has also been highlighted in patients with HIV infection. In these patients, the percentage of IL-17-producing cells is significantly higher than in non-infected subjects, with IL-17 level positively correlating with markers of neutrophil activation, another typical feature of HIV pathology [99].

Taken together, numerous studies show that IL-17 is capable of inducing sustained recruitment and activation of neutrophils during infections involving several types of viruses. The blockade of IL-17 signalling results in alleviated tissue damage during virus infection in murine models. In the light of these findings, it would be of interest to study whether anti-IL-17 therapy could potentially serve as a targeted strategy for treatment of such viral infection cases, especially in very severe cases. This is likely to be challenging work due to the pleiotropic role of IL-17.

### 3.2.2. Promoting fibrosis development

Fibrosis is a consequence of chronic injury of a target tissue characterized by excessive deposition of extracellular matrix (ECM) proteins and a typical feature of many chronic diseases [100,101]. The progression of fibrosis usually results in the distortion of normal target tissue architecture via pathologic tissue remodelling and fibrous scar formation. This in turn creates fundamental changes in organ function and can even result in organ failure [100,101]. Viruses may induce tissue damage by promoting fibrosis development, as is particularly true for chronic HBV or HCV infections, whereby inappropriate activation of IL-17-producing cells plays a critical role in maintaining fibrogenic pathways and disease progression. Abundant evidence suggests a potential correlation between IL-17-producing cells (i.e. Tc17 cells or Th17 cells) or IL-17 levels and a more severe disease stage of HBV or HCV progression [102–106], while mechanistic studies have shown a causal link between IL-17 production and the development of liver fibrosis. For example, IL-17 expression has been observed to be specifically localized to portal areas and fibrotic septa in patients with HBV or HCV infection [107]. Interestingly, most of these IL-17-expressing cells are neutrophils, with the number of these cells showing a strong positive correlation with liver fibrosis stage [107]. More specifically, the work of Wang *et al.* [108] has highlighted a critical role of the IL-23/IL-17 axis in the induction of liver damage after HBV infection. In HBV-infected patients, IL-23 production by DCs and macrophages stimulates the differentiation of naive CD4$^+$ T cells into Th17 cells, the primary source of IL-17 in HBV-infected livers. At the same time, IL-17R is highly expressed on hepatic stellate cells (HSCs), one of the most important collagen-producing cell types in the liver (figure 2). Another study found that the production of Th17 differentiation cytokines (i.e. TGF-β,

royalsocietypublishing.org/journal/rsob   Open Biol. 9: 190109

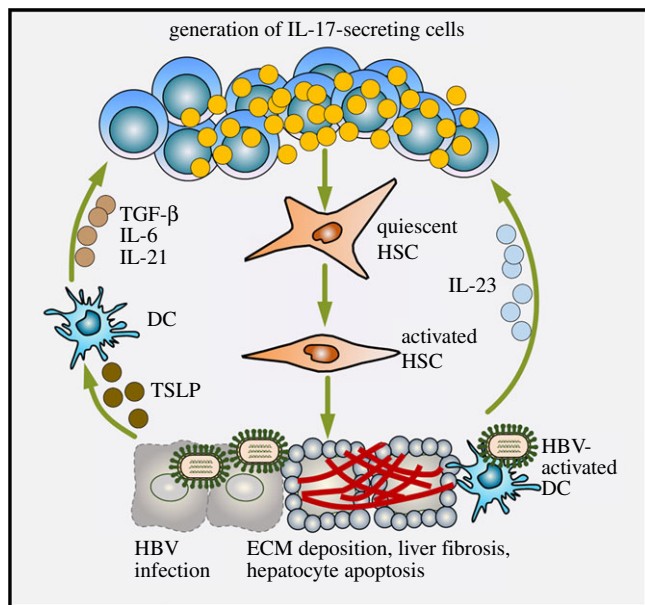

**Figure 2.** IL-17 promotes the progression of liver fibrosis during HBV infection. During HBV infection, virus-infected hepatocytes release TSLP, which stimulates the production of Th17 differentiation-associated cytokines (i.e. TGF-β, IL-6 and IL-21) from hepatic DCs, leading to the differentiation of naive CD4$^+$ T cells into IL-17-producing Th17 cells. In addition, virus-activated DCs also stimulate the generation of Th17 cells by producing IL-23. The production of IL-17 activates HSCs, the major ECM-producing cells in the liver. Excessive ECM degradation leads to liver fibrosis and hepatocyte apoptosis observed in these patients. TGF-β, transforming growth factor-β; DC, dendritic cell; TSLP, thymic stromal lymphopoietin; HSC, hepatic stellate cell; HBV, hepatitis B virus; ECM, extracellular matrix.

IL-6 and IL-21) by hepatic DCs could also be stimulated by thymic stromal lymphopoietin (TSLP) released from HCV-infected hepatocytes, with substantial levels of TSLP detected in fibrotic liver tissues of patients with chronic HCV [109] (figure 2).

Moore and colleagues [110,111] used a murine syngeneic bone marrow transplantation model to highlight the essential role of IL-17 in inducing pneumonitis and pulmonary fibrosis during gammaherpesvirus infection. Their results demonstrated that mice after syngeneic bone marrow reconstitution and concurrent gammaherpesvirus infection exhibited skewed differentiation of CD4$^+$ T cells into Th17 cells. By producing IL-17A, these Th17 cells directly induced the activation and production of ECM by lung mesenchymal cells [110]. Mechanistically, deficiency of Notch ligand DLL4 in lung CD103$^+$ and CD11b$^+$ DCs was shown to be responsible for the pathologic Th17 responses [111]. Notably, restored Notch signalling or IL-17 neutralization attenuated fibrosis and pneumonitis in these mice [110,111].

The body of evidence above suggests that during infections by certain viruses, IL-17 promotes fibrosis development. However, most of these results are based on murine studies or *in vitro* human studies. More relevant human-based *in vivo* evidence and additional in-depth mechanistic investigations are needed to elucidate how IL-17 drives fibrosis development after viral infections.

### 3.2.3. Antagonizing development of Treg cells

Upon activation, naive CD4$^+$ T cells can differentiate into a variety of effector cell subsets, including Th1, Th2, Th17, follicular helper T cells, induced regulatory T (iTreg) cells and others [112]. The lineage determination of naive CD4$^+$ T cells is governed predominantly by the cytokines present in the microenvironment [113]. Intriguingly, transforming growth factor (TGF)-β is not only indispensable for the development of iTregs but is also a potent inducer of Th17 cells, with the balance between Treg and Th17 cell differentiation depending on the overall cytokine milieu. More specifically, a lower concentration of TGF-β in the presence of several other proinflammatory cytokines, such as IL-1β, IL-6, IL-21 and IL-23, may promote a Th17 response, while a higher concentration in combination with IL-2 appears to promote an iTreg response [114,115]. Therefore, the abnormal expansion of Th17 cells may potentially antagonize the development of Treg cells, resulting in insufficient immune regulation, persistent immune activation and consequently greater degrees of immunopathology, all of which are particularly relevant to several types of viral infections.

Patients with chronic hepatitis B (CHB) showed that Th17 cell percentage is negatively correlated with Treg frequencies, aligning with observations that Th17 cells and Treg cells usually show distinct and opposing features that have clinical relevance to disease severity [116–118]. Th17 frequency has been shown to positively correlate with levels of liver stiffness, serum levels of alanine aminotransferase, total bilirubin levels and prothrombin time, while Treg cell frequency negatively correlated with liver stiffness [117,119,120]. Importantly, an increased Th17/Treg ratio has frequently been reported in CHB patients, and this ratio positively correlates with liver stiffness level and extent of liver injury, suggesting a potential link between Th17/Treg balance and liver cirrhosis progression in those patients [118,119,121].

An imbalance between Th17 and Treg cells has also been reported in cases of viral infections of the respiratory tract. Qin *et al.* [122] found that after RSV infection, human bronchial epithelial cells potently induced the differentiation of Th17 cells, while inhibiting the differentiation of Treg cells. Consistent with this finding, Gao *et al.* [123] using a rat model of RSV bronchiolitis observed higher Th17 cell percentages and IL-17 and IL-23 concentrations but significantly reduced overall Treg cell percentages and IL-10 levels. In other work, Li *et al.* observed a significantly reduced percentage of Treg cells in children suffering from RSV infection, with concomitantly decreased IL-10 levels compared with healthy controls. In addition, the concentration of IL-17 and the frequency of Th17 cells were significantly increased in these patients [124]. When galectin-9, a specific inhibitor of Th1 and Th17 immune responses, was administered to RSV-infected mice, significantly decreased viral load and diminished lung pathology were observed that mainly resulted from the suppression of both Th17 differentiation and the induction of Treg cell expansion [125]. Similarly, in a murine model of human metapneumovirus infection, *IL-17* deletion resulted in an increased percentage of lung-infiltrating Treg cells and reduction in Th1 and Th2 cells in the lung [93].

Collectively, these findings show that an imbalance between Th17 and Treg cells might be a critical pathogenic factor dictating clinical severity of patients with certain types of viral infection. Therefore, such an imbalance can serve as an important indicator for use in clinical diagnosis and for evaluating treatment efficacy in these patients.

royalsocietypublishing.org/journal/rsob Open Biol. 9: 190109

Moreover, therapies targeted at restoring a normal Th17/Treg ratio may hold substantial promise for such patients in the future.

### 3.2.4. Inducing Th2 immune responses

Patients with RSV infection usually develop severe bronchiolitis that is ascribed to a potent Th2 immune response in the respiratory tract. This notion is supported by the finding that IL-4 and IL-4 receptor α polymorphisms are associated with severe RSV bronchiolitis [126], and high levels of both Th2-associated transcriptional factors and cytokines have been detected in patients with more severe manifestations [127–131]. Mounting evidence has shown that IL-17A can induce the differentiation of Th2 cells [132,133], as is observed in RSV infections. Several murine models have shown that IL-17 and IL-13 are usually simultaneously produced after RSV infection, implicating a concurrent Th17 and Th2 response. The underlying molecular mechanisms have implicated a critical role for the IL-27/STAT1 (signal transducer and activator of transcription 1) signalling pathway in enhancing Th17 and Th2 responses, as both IL-27 receptor-deficient and STAT1-deficient mice exhibited significantly elevated levels of IL-17 and Th2-associated cytokines [92,134]. Importantly, IL-17 neutralization in both mouse models resulted in improved disease outcomes and reduced Th2 cytokine production, as observed in several other independent studies [77,122,135]. Intriguingly, Monick *et al.* [136] observed that IL-4 and IL-13 could potently stimulate the production of Th2 cell chemoattractant CCL17 by lung epithelial cells after RSV infection, suggesting a positive feedback loop for Th2 cell generation after RSV infection.

Although published reports have implicated an IL-17-induced Th2 immune response in mediating airway dysfunction after RSV infection, most of the findings are based on animal models or mere observational patient data. Further work to elucidate the involvement of IL-17 and its relevant mechanism of action in regulation of Th2 cells is needed. Considering that Th2-targeted therapies have been shown to be effective for treatment of asthma [137,138], a common subsequent complication of RSV infection [139], the findings above suggest that Th2 immune response-based therapies may be promising treatments for RSV-induced bronchiolitis.

## 4. Why IL-17 plays so many diverse roles in viral infections

As discussed above, IL-17 is a pleiotropic cytokine that can drive pathogenicity in multiple tissues, fine-tune inflammatory responses and maintain tissue integrity during viral infections. However, it is not well understood why IL-17 has so many diverse functions in viral infections. We propose that in addition to common reasons such as viral types, stains and infection doses, several additional explanations may also warrant consideration.

Although IL-17R is expressed universally in the body, the subsequent signalling pathway mediated by IL-17/IL-17R relies greatly on both the specific cell types present and on the tissue microenvironment. For example, IL-17 stimulates the expression of *Csf2* (encoding granulocyte-macrophage colony-stimulating factor (GM-CSF)) in NK cells to foster the differentiation and proliferation of Kupffer cells [74,75],

thus endowing this cell subset with competence to control systemic candidiasis. Alternatively, in lung epithelium, IL-17/IL-17R signalling stimulates the activation of extracellular signal-regulated kinase 1/2 and induces the expression of *IL-8*, *IL-6* and *CXCL5* to facilitate neutrophil recruitment [140,141]. In yet another scenario involving intestinal epithelial cells, IL-17 signalling through an Act-1 pathway induces the production of occludin [76] and mucin [142], which are crucial for maintenance of gut tissue integrity. Moreover, in the liver IL-17 can directly activate the STAT3 signalling of HSCs to induce the production of collagen type I, thus contributing to hepatic fibrosis [74], while also exhibiting direct mitogenic stimulation of hepatocytes and liver progenitor cells to promote their proliferation [143,144]. However, despite their IL-17R expression, liver endothelial cells do not respond to IL-17 stimulation [74]. Moreover, IL-17 can stimulate the proliferation and osteoblastic differentiation of bone mesenchymal progenitor cells to promote bone formation after injury [145]. Cell type- and tissue microenvironment-dependent IL-17/IL-17R signalling pathways may hold clues to understanding the diverse functions of IL-17.

In most viral infections, tissue inflammation is not driven by IL-17 alone, but by the concerted actions of IL-17 and many other proinflammatory and immunoregulatory signals. Thus, the final immunological response and subsequent severity of viral infections are highly dependent on the collective effect of these signals, a view that may explain the protective and pathogenic functions of IL-17 in different settings of inflammation. Th17 and Treg cells are both activated and function in opposition regarding RSV clinical pathology and viral clearance. Th17 cells have been reported to stimulate the chemotaxis of neutrophils into the infected lungs [77,90,91], induce a Th2 skewed immune response [77,122,135] and dampen CTL activity through negative regulation of effector cytokines and cytolytic-associated gene expression [77]. Conversely, Treg cells are involved in both inhibition of excessive immune responses and promotion of the early recruitment of virus-specific CD8$^+$ T cells [146,147]. Thus, a delicate balance of Th17/Treg ratio is crucial to fine-tune the inflammatory response in the lung after RSV infection. If this balance is maintained properly, the pathologic effects of IL-17 can be held in check, and its beneficial functions (i.e. maintaining the integrity of lung epithelium) will far outweigh its pathogenic effects. In fact, in addition to RSV infections, this phenomenon has also been demonstrated in many other viral and non-viral infections [7,148,149] and even in several non-infectious diseases [150].

Viruses may simultaneously activate multiple IL-17-producing cell subsets that differ in several key biological activities. For example, Th17 cells are very permissive to HIV infection and can promote the intracellular replication of HIV, such that the presence of these cells correlates well with HIV pathology [39]. On the other hand, Tc17 cells, which show no such attributes and which express FasL, are involved in inhibiting the pathogenic inflammation brought about by HIV infection [55,66,79]. Similarly, following influenza infection of the lung, the presence of Th17 cells exacerbates pathology, while the number of Tc17 cells is negatively associated with morbidity and mortality [53,95]. In comparison, IL-17A-producing $\gamma\delta$T cells in the lung can upregulate expression of IL-33 and amphiregulin, thereby

royalsocietypublishing.org/journal/rsob    Open Biol. 9: 190109

promoting lung repair following influenza infection [73]. Thus, the diverse functions of IL-17 in viral infections may be attributed to the unique effector functions of different IL-17-producing cell subsets.

## 5. Therapeutic potential of targeting IL-17 during viral infections

Taking into consideration that IL-17 actively participates in various types of viral infections, agents that can modulate IL-17, IL-17-associated cytokines and IL-17-producing cells hold promise for suppressing viral infections and minimizing tissue pathology. In fact, some of these agents have already yielded benefits in several preliminary investigations.

Leflunomide is an immunosuppressive drug that inhibits mitochondrial dihydroorotate dehydrogenase, blocking de novo pyrimidine synthesis and cell proliferation particularly in activated lymphocytes [3,142]. In addition, higher concentrations of leflunomide may also inhibit protein kinase activity and the NF-κB signalling pathway in B and T lymphocytes [4,151]. In fact, leflunomide has shown effective antiviral activity towards cytomegalovirus [7], herpesvirus [8,10] and HIV-1 [9] and has also been used for treatment of polyomavirus-associated nephropathy [15]. Although the detailed mechanisms underlying its antiviral effects are not completely understood, accumulating evidence indicates that suppression of IL-17 production may be partly involved. For instance, treatment of peripheral blood lymphocytes with leflunomide resulted in the activation of these cells that was accompanied by decreased production of IL-17 and tumour necrosis factor-α, two cytokines that usually functionally synergize to initiate proinflammatory responses [11]. In addition, A771726, an active leflunomide metabolite, significantly inhibited the production of IL-1β and IL-6, two crucial stimulators for the differentiation of IL-17-producing cells [13]. Meanwhile, murine-based *in vivo* evidence has confirmed that A771726 treatment resulted in attenuated STAT3 activity in CD4$^+$ T cells and reduced Th17 cell numbers [14]. Until now, the antiviral effect of leflunomide has not been evaluated in clinical trials in patients with viral infections, but its suppressive effect on IL-17 production has already been tested in patients with rheumatoid arthritis [152]. Notably, leflunomide–methotrexate combination therapy in these patients significantly decreased circulating Th17 cells and the plasma levels of IL-17 and was associated with ameliorated RA symptoms [152].

In patients with chronic HBV or HCV, IL-17 production is essentially involved in the initiation and progression of sustained liver inflammation and fibrosis. A wide range of agents has been tested for therapeutic efficacy in treating fibrosis by targeting IL-17 in these patients. Vitamin D has been shown to suppress the differentiation and expansion of Th17 cells and to promote Treg cell differentiation [153]. In patients with chronic HBV or HCV, vitamin D level negatively correlated with the severity of liver fibrosis, viral load and adverse clinical outcomes [16–18,154]. In patients with chronic HCV or decompensated liver cirrhosis, vitamin D supplementation significantly alleviated liver fibrosis or cirrhosis, improved disease progression and resulted in higher survival rates [155–157]. Rapamycin is another agent that reduces liver fibrosis by antagonizing the effect of IL-17 through possible abrogation of the IL-6-induced Th17

response during acute-on-CHB liver failure [158]. Meanwhile, other studies have shown that bone marrow-derived mesenchymal stem cell transplantation also shows efficacy in ameliorating liver fibrosis and improving liver function in patients with HBV-related liver cirrhosis, with the modulation of Treg/Th17 balance partly accounting for the beneficial effect in these patients [159,160].

As discussed above, IL-17 may suppress certain viral infections and infection-associated tissue injuries, leading one to assume that the enhancement of IL-17 activities may confer antiviral effects. However, no clinical trials of therapies have yet been reported and we suggest that clinical trials may not be viable in most cases. Major concerns stem from the fact that excessive IL-17 and IL-17-producing cells can induce severe inflammatory responses and associated tissue damage, such that the side effects of this approach would far outweigh its potential benefits. A more gentle induction of IL-17 without inducing unnecessary immune responses will be sought in the future to mitigate these effects.

## 6. Conclusion and future perspectives

We have discussed current understanding regarding the complex functions of IL-17 in viral infections and the underlying mechanisms. We have also described the translational potential of IL-17-targeted therapeutics for treating viral infection-associated illnesses. Although the role of IL-17 has been extensively investigated in a wide range of animal models and clinical trials, several intriguing aspects of this cytokine warrant further investigations. For example, while IL-17 can be produced by cells of both the innate and adaptive immune system, the functional relationships between these cells are not clear. In addition, in view of the importance of vaccination for the prevention of viral infections, future work investigating IL-17 and IL-17-secreting cells in virus rechallenge models are needed to gain a better understanding of relevant memory (or memory-like) responses.

Specific therapies for viral infections and related diseases remain elusive for many viruses. A better understanding of the underlying mechanisms of the antiviral immune responses may lead to novel treatment options targeted at suppressing pathogenic immunity while improving beneficial immunity. As discussed above, it appears that the functions of IL-17 are crucial and complex in different settings of viral infections. These features make it difficult to evaluate whether the benefits of a given targeted therapy would outweigh the potential risks, thus greatly hindering the application of IL-17-targeted therapies. However, in certain cases, targeting of the downstream signalling in lieu of direct manipulation of IL-17 may avoid unnecessary side effects and serve as an effective alternative therapeutic strategy in the future [155]. In addition, considering that the transduction of IL-17 signalling pathway and activation of the IL-17-secreting cells are controlled by sophisticated regulatory mechanisms [154], a better understanding of the relevant mechanisms may also shed light on strategies that optimally exploit the use of IL-17.

In conclusion, in the case of West Nile virus, adenovirus and vaccinia virus infection, IL-17 plays a positive role in antiviral immune responses. However, in the case of Theiler's murine encephalomyelitis virus, Coxsackievirus, dengue virus, HBV, HCV and gammaherpesvirus infection, IL-17

can promote and exacerbate virus-induced illnesses. In addition, during influenza virus, HSV, RSV, SIV and HIV infection, IL-17 can play both protective and pathogenic roles. Several agents that can downregulate IL-17 activities can effectively suppress certain types of viral infections and limit tissue pathology. However, a great challenge that needs to be considered and overcome is that the enhancement of IL-17 activities is not feasible in cases of viral infections due to severe side effects. Therefore, investigations into identifying alternative methods, such as interfering with downstream IL-17 signalling pathway, are urgently needed in the future.

Data accessibility. This article has no additional data.

Authors' contributions. W.-T.M. and D.-K.C. drafted the original manuscript; X.-T.Y., Q.P. and D.-K.C. critically revised the manuscript. All authors gave final approval for publication and agree to be held accountable for the work performed therein.

Competing interests. The authors declare that no conflict of interest exists.

Funding. This work was supported by the Key Industrial Innovation Chains of Shaanxi Province (2018ZDCXL-NY-01-06), Qinghai Province Major R&D and Transformation Project (2018-NK-125), Xianyang Science and Technology Major Project (2017K01-34), the Youth Innovation Team of Shaanxi Universities and PhD Research Startup Fund of Northwest Agriculture and Forestry University (00500/Z109021716).

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
