## [Reviewer comments · Open Biology]

Review History

RSOB-19-0109.R0 (Original submission)

Review form: Reviewer 1

Recommendation

Accept with minor revision (please list in comments)

Are each of the following suitable for general readers?

- a) **Title**
Yes
- b) **Summary**
Yes
- c) **Introduction**
Yes

Is the length of the paper justified?

Yes

Should the paper be seen by a specialist statistical reviewer?

No

Is it clear how to make all supporting data available?

Is the supplementary material necessary; and if so is it adequate and clear?

Do you have any ethical concerns with this paper?

No

Comments to the Author

Comments for the Authors

This is a very well written, easy to follow review discussing the critical roles of IL-17 in antiviral immune responses and virus-induced illnesses. The authors provide the appropriate amount of background and have done a nice job of pointing out the necessity of IL-17-involved therapeutic strategies, which are uniquely tailored to both the infectious agent and the infection context. The figures are nicely drawn and help to illustrate understudied areas. While the review is very clear and fairly comprehensive, there are a few suggestions that the authors may wish to consider to even further strengthen the review.

1. In a number of places, statements are made that are not supported by references. It would be helpful for those readers not in the field to include references for these statements. Specific examples include but are not limited to: in section 1.2(Peng et al.) can also be cited in 'Enhancing the survival of virus-infected cells' section to illustrate IL - 17 is able to enhance the survival of the virus infected cells, but it does not mean that this kind of surviving virus-infected cell would promote viral infections and mediates viral infection-induced pathology.

2. The author put forward two different arguments in the form of questions in the title, but there are only vague notions in the conclusion. They don't answer the questions that raised at the beginning of the paper. Perhaps the authors should make it clearly by using 'In the case..., IL-17 plays a positive role in antiviral immune responses, while in the case..., IL-17 can promote and exacerbate virus-induced illnesses.' What challenges might need to be considered and/or overcome? Some discussion on these points would help to synthesize the data presented in the review and distill it down to a take home message for the readers.

3. There are many grammatical errors throughout the manuscript: no "the" before some nouns which followed by "of", such as "influenza infection of neonatal mice", "activation of the IL-17-secreting cells..."; "Further investigation revealed a loss of IL-17-producing cells due to depletion of CD103+ DCs, which have been shown to both express genes favorable to IL-17 production and induce the differentiation of IL-17A- and RORc-expressing cells upon coculture with naive T cells in vitro"; "During murine systemic infection by HSV, Stout-Delgado et al. found that the increased morbidity and mortality of aged mice was caused by a rapid increase in IL-17 level (primarily produced by hepatic NKT cells) during infection"

Decision letter (RSOB-19-0109.R0)

24-Jun-2019

Dear Dr Ma

We are pleased to inform you that your manuscript RSOB-19-0109 entitled "The protective and pathogenic roles of IL-17 in viral infections: friend or foe?" has been accepted by the Editor for

publication in Open Biology. The reviewer has recommended publication, but also suggest some minor revisions to your manuscript. Therefore, we invite you to respond to the reviewer's comments and revise your manuscript.

Please submit the revised version of your manuscript within 7 days. If you do not think you will be able to meet this date please let us know immediately and we can extend this deadline for you.

- 1) A text file of the manuscript (doc, txt, rtf or tex), including the references, tables (including captions) and figure captions. Please remove any tracked changes from the text before submission. PDF files are not an accepted format for the "Main Document".
- 2) A separate electronic file of each figure (tiff, EPS or print-quality PDF preferred). The format should be produced directly from original creation package, or original software format. Please note that PowerPoint files are not accepted.
- 3) Electronic supplementary material: this should be contained in a separate file from the main text and meet our ESM criteria (see <http://royalsocietypublishing.org/instructions-authors#question5>). All supplementary materials accompanying an accepted article will be treated as in their final form. They will be published alongside the paper on the journal website and posted on the online figshare repository. Files on figshare will be made available approximately one week before the accompanying article so that the supplementary material can be attributed a unique DOI.

Online supplementary material will also carry the title and description provided during submission, so please ensure these are accurate and informative. Note that the Royal Society will not edit or typeset supplementary material and it will be hosted as provided. Please ensure that the supplementary material includes the paper details (authors, title, journal name, article DOI). Your article DOI will be 10.1098/rsob.2016[last 4 digits of e.g. 10.1098/rsob.20160049].

- 4) A media summary: a short non-technical summary (up to 100 words) of the key findings/importance of your manuscript. Please try to write in simple English, avoid jargon, explain the importance of the topic, outline the main implications and describe why this topic is newsworthy.

Images

Data-Sharing

It is a condition of publication that data supporting your paper are made available. Data should be made available either in the electronic supplementary material or through an appropriate repository. Details of how to access data should be included in your paper. Please see <http://royalsocietypublishing.org/site/authors/policy.xhtml#question6> for more details.

Data accessibility section

Sincerely,

The Open Biology Team

<mailto:openbiology@royalsociety.org>

Reviewer(s)' Comments to Author:

Referee: 1

Comments to the Author(s)

Comments for the Authors

This is a very well written, easy to follow review discussing the critical roles of IL-17 in antiviral immune responses and virus-induced illnesses. The authors provide the appropriate amount of background and have done a nice job of pointing out the necessity of IL-17-involved therapeutic strategies, which are uniquely tailored to both the infectious agent and the infection context. The figures are nicely drawn and help to illustrate understudied areas. While the review is very clear and fairly comprehensive, there are a few suggestions that the authors may wish to consider to even further strengthen the review.

1. In a number of places, statements are made that are not supported by references. It would be helpful for those readers not in the field to include references for these statements. Specific examples include but are not limited to: in section 1.2(Peng et al.) can also be cited in 'Enhancing the survival of virus-infected cells' section to illustrate IL - 17 is able to enhance the survival of the virus infected cells, but it does not mean that this kind of surviving virus-infected cell would promote viral infections and mediates viral infection-induced pathology.
2. The author put forward two different arguments in the form of questions in the title, but there are only vague notions in the conclusion. They don't answer the questions that raised at the beginning of the paper. Perhaps the authors should make it clearly by using 'In the case..., IL-17 plays a positive role in antiviral immune responses, while in the case..., IL-17 can promote and exacerbate virus-induced illnesses.' What challenges might need to be considered and/or

overcome? Some discussion on these points would help to synthesize the data presented in the review and distill it down to a take home message for the readers.

3. There are many grammatical errors throughout the manuscript: no “the” before some nouns which followed by “of”, such as “influenza infection of neonatal mice”, “activation of the IL-17-secreting cells...”; “Further investigation revealed a loss of IL-17-producing cells due to depletion of CD103+ DCs, which have been shown to both express genes favorable to IL-17 production and induce the differentiation of IL-17A- and RORc-expressing cells upon coculture with naive T cells in vitro”; “During murine systemic infection by HSV, Stout-Delgado et al. found that the increased morbidity and mortality of aged mice was caused by a rapid increase in IL-17 level (primarily produced by hepatic NKT cells) during infection”

Author's Response to Decision Letter for (RSOB-19-0109.R0)

See Appendix A.

Decision letter (RSOB-19-0109.R1)

02-Jul-2019

Dear Dr Ma

We are pleased to inform you that your manuscript entitled "The protective and pathogenic roles of IL-17 in viral infections: friend or foe?" has been accepted by the Editor for publication in Open Biology.

Article processing charge

Please note that the article processing charge is immediately payable. A separate email will be sent out shortly to confirm the charge due. The preferred payment method is by credit card; however, other payment options are available.

Sincerely,

The Open Biology Team
mailto: openbiology@royalsociety.org

Appendix A

Response to referees:

We would like to thank the editorial office and the referee for their comprehensive assessment of our manuscript. We have taken all the comments into account, in a manner that is described in detail below. The contents that have been revised were marked bold and underlined in the main manuscript. We think that, following the insightful suggestions of the editorial office and the reviewer, our manuscript has gained in clarity and hope that the changes made will be considered satisfactory.

Referee: 1

This is a very well written, easy to follow review discussing the critical roles of IL-17 in antiviral immune responses and virus-induced illnesses. The authors provide the appropriate amount of background and have done a nice job of pointing out the necessity of IL-17-involved therapeutic strategies, which are uniquely tailored to both the infectious agent and the infection context. The figures are nicely drawn and help to illustrate understudied areas. While the review is very clear and fairly comprehensive, there are a few suggestions that the authors may wish to consider to even further strengthen the review.

1. In a number of places, statements are made that are not supported by references. It would be helpful for those readers not in the field to include references for these statements. Specific examples include but are not limited to: in section 1.2(Peng et al.) can also be cited in 'Enhancing the survival of virus-infected cells' section to illustrate IL-17 is able to enhance the survival of the virus infected cells, but it does not mean that this kind of surviving virus-infected cell would promote viral infections and mediates viral infection-induced pathology.

Response: Thank you for your kind advice. We have revised the manuscript to avoid such misunderstandings. To be specific, we have revised the sentence "Inducing neutrophil migration and activation" as "Inducing excessive neutrophil migration and activation" at line 392 to indicate a pathogenic role of IL-17, because IL-17-induced neutrophil migration at a moderate level participates in protective antiviral immune responses as discussed at lines 208-213. In addition, we have moved the content discussing the finding by Peng et al. into section 2.1 (Enhancing the survival of virus-infected cells). By doing so, this finding can be served as a special example showing the promotion of virus permissive cells by IL-17 does not always promote viral infections, so as to avoid the misunderstanding you worried. Please see the new manuscript at lines 353-361.

2. The author put forward two different arguments in the form of questions in the title, but there are only vague notions in the conclusion. They don't answer

the questions that raised at the beginning of the paper. Perhaps the authors should make it clearly by using 'In the case..., IL-17 plays a positive role in antiviral immune responses, while in the case..., IL-17 can promote and exacerbate virus-induced illnesses.' What challenges might need to be considered and/or overcome? Some discussion on these points would help to synthesize the data presented in the review and distill it down to a take home message for the readers.

Response: Thanks for your insightful suggestion. We have answered the questions that raised at the beginning of the paper and discussed what challenges might need to be considered and/or overcome. Please see the new manuscript at lines 713-724.

3. There are many grammatical errors throughout the manuscript: no "the" before some nouns which followed by "of", such as "influenza infection of neonatal mice", "activation of the IL-17-secreting cells..."; "Further investigation revealed a loss of IL-17-producing cells due to depletion of CD103+ DCs, which have been shown to both express genes favorable to IL-17 production and induce the differentiation of IL-17A- and RORc-expressing cells upon coculture with naive T cells in vitro"; "During murine systemic infection by HSV, Stout-Delgado et al. found that the increased morbidity and mortality of aged mice was caused by a rapid increase in IL-17 level (primarily produced by hepatic NKT cells) during infection"

Response: Thank you very much for your comment and we are sorry for our mistakes. We have critically revised the manuscript. Please see the relevant revisions throughout the new manuscript.